# Substituting Inorganic Fertilizers with Organic Amendment Reduced Nitrous Oxide Emissions by Affecting Nitrifiers' Microbial Community

**Lihua Xie [1], Lingling Li [1],\*, Junhong Xie [1], Jinbin Wang [1], Sumera Anwar [2], Changliang Du [1] and Yongjie Zhou [1]**

[1] State Key Laboratory of Aridland Crop Science, College of Agronomy, Gansu Agricultural University, Lanzhou 730070, China

[2] Institute of Molecular Biology and Biotechnology, The University of Lahore, Lahore 54660, Pakistan

\* Correspondence: lill@gsau.edu.cn

**Abstract:** Excessive inorganic fertilizers are one of the main causes of nitrous oxide ($N_2O$) emissions. Organic fertilizers can not only reduce the use of nitrogen (N) fertilizers by increasing soil organic matter but are also safe for the environment. The partial replacement of nitrogen (N) fertilizers with organic fertilizers can potentially reduce $N_2O$ emissions. To illuminate the best ratio for the nitrogen replacement of inorganic fertilizer, the present experiment was conducted in dryland areas of central Gansu Province and different portions of inorganic N fertilizers (200 kg ha$^{-1}$); i.e., 0, 50, 37.5, 25, and 12.5% were replaced with commercial organic fertilizers to test their effects on soil physicochemical properties, the grain yield of maize, $N_2O$ emissions, and the diversity of ammonia-oxidizing archaea (AOA) and bacterial (AOB) communities. Results showed that the maximum $N_2O$ emission was obtained by 100% inorganic fertilizers and the lowest was obtained at the control (no fertilizer). Substituting inorganic fertilizers with organic manure not only reduced $N_2O$ emissions but also improved soil organic carbon content and soil moisture and typically improves grain yield and biomass. The highest reduction in $N_2O$ emissions was recorded by 50% substitution. Furthermore, 37.5% and 12.5% substitutions did not reduce the grain yield and biomass compared to 100% inorganic fertilizer, and a 37.5% substitution performed better in improving soil fertility. Organic fertilizer increased the amoA copy number of AOA but decreased that of AOB. Nitrososphaera (AOA) and Nitrosospira (AOB) were the most dominant ammonia-oxidizing communities. Structural equation modeling indicated that AOB contributes more $N_2O$ emissions than AOA and is more sensitive to changes in pH, moisture, and $NO_3^- - N$, and the input of organic fertilizers may affect AOB by influencing soil physicochemical traits. In summary, replacing a reasonable proportion (37.5%) of inorganic fertilizers with organic manure improves soil fertility, reduces $N_2O$ emissions, and stabilizes production.

**Keywords:** soil properties; nitrification-related $N_2O$ emission; organic fertilizer; ammonia oxidizer

## 1. Introduction

The emission of greenhouse gases (GHG) is a serious ecological problem and one of the basic reasons for global warming, as they contribute 44% of global warming by trapping radiation [1,2]. Semi-arid regions have faced the most significant global warming effects over the last 100 years [3,4]). Since 1990, emissions of greenhouse gases from agriculture increased by 12%, with a 60% increase in emissions of $N_2O$ and $CH_4$ [5]. Agriculture is a source of almost 10% of GHG emissions, with a total of 6558 Mt $CO_2$ equivalent [6]. Nitrous oxide ($N_2O$) depletes the ozone layer; moreover, it potentially is the most damaging greenhouse gas, as the impact of $N_2O$ emissions is 300 times greater than $CO_2$ [7–9]. Hence, the focus of this study is $N_2O$ emissions and its mitigation.

Almost 80% of $N_2O$ is emitted from agricultural activities [10]. The high use of N fertilizers is mainly responsible for $N_2O$ emissions, primarily through nitrification and denitrification processes. The first step of nitrification is ammonia oxidation, carried out by ammonia mono-oxygenase (AMO), a rate-limiting enzyme that determines the rate of the ammonia oxidation process [11–13]. Ammonia oxidation is thought to be carried out by ammonia oxidizers, such as ammonia-oxidizing archaea (AOA) and ammonia-oxidizing bacteria (AOB). The nitrification rate and communities of these groups are typically identified using the strongly conserved marker amoA gene, which encodes the alpha subunit of AMO.

Fertilizers affect the abundance, richness, and community composition of nitrifying bacteria by influencing the edaphic environment, i.e., pH, nitrate–nitrogen, ammonium–nitrogen, organic matter, and moisture, which are coupled with $N_2O$ emissions [14]. The magnitude and contribution of fertilizers and soil biochemical properties to $N_2O$ emissions vary with agroecosystems [15–17].

Many studies aiming to reduce GHS emissions have been conducted in the semi-arid Loess Plateau region in Gansu Province, China [18,19], focusing on yield, nitrogen loss, and soil quality [20,21]. These reports showed that organic fertilizers effectively reduced $N_2O$ emissions from farmland, but there is no consensus on how it influences AOA and AOB communities. The relative contribution of AOA and AOB to $N_2O$ emissions during ammonia oxidation is poorly appreciated with the incorporation of different rates of organic fertilizer.

This study is intended to investigate the effects of substituting different portions of inorganic fertilizers with organic fertilizer on $N_2O$ emissions from maize fields. It was hypothesized that $N_2O$ emissions could be reduced by using different proportions of inorganic and organic fertilizers, which would influence the abundance and activities of AOA and AOB.

## 2. Materials and Methods

### 2.1. Overview of the Experimental Area

The experimental station (35°28′ N, 104°44′ E) is located in the semiarid zone of the western Loess Plateau, Gansu province of China. The average daily temperature is 6.4 °C, the average sunshine is 2477 h, and the annual solar radiation is 593 kJ $cm^{-2}$. In 2020 and 2021, rainfalls were 524.0 and 317.3 mm, respectively. The soil at the site is a Calcareous Cambisol [22]. The initial soil physiochemical properties at the experimental site are shown in Table 1.

**Table 1.** Initial physiochemical properties in the soil before the experimental setup.

| Soil Depth (cm) | Bulk Density (g $cm^{-3}$) | pH | Total Nitrogen (g $kg^{-1}$) | Total Phosphorus (g $kg^{-1}$) | Organic C (g $kg^{-1}$) |
|---|---|---|---|---|---|
| 0–5 | 1.24 | 8.37 | 0.91 | 1.04 | 8.80 |
| 5–10 | 1.24 | 8.43 | 0.99 | 1.10 | 8.67 |
| 10–30 | 1.35 | 8.49 | 0.83 | 1.07 | 8.48 |

Values are means ($n$ = 3).

### 2.2. Experimental Design

From 2016 to 2021, different combinations of organic and inorganic fertilizers were applied to rainfed summer maize (*Zea mays* L. variety Xianyu 335) in a randomized block design with three replications. Different portions of inorganic fertilizer, i.e., 0 (T1), 50.0% (T2), 37.5% (T3), 25.0% (T4), and 12.5% (T5) were replaced with organic fertilizers, while no fertilizer was applied to the control (T6). T1 comprised 100% inorganic nitrogen fertilizer for which urea (46% N) was used, and $\frac{1}{2}$ of the total urea was applied before sowing, and the other $\frac{1}{2}$ was applied in the proportion of 3:2 at the jointing stage and the trumpet period (Table 2).

**Table 2.** Different treatments of nitrogen fertilizers.

| Treatment | Substitution % | Total Fertilizer Rate | Base Fertilizer | Topdressing (Jointing Stage) | Topdressing (Pre-Tasseling) |
|---|---|---|---|---|---|
| T1 | 0% | Inorganic 200 kg N ha$^{-1}$ | Inorganic 100 kg N ha$^{-1}$ | Inorganic 60 kg N ha$^{-1}$ | Inorganic 40 kg N ha$^{-1}$ |
| T2 | 50.0% | Organic 100 kg N ha$^{-1}$ + Inorganic 100 kg N ha$^{-1}$ | Organic 100 kg N ha$^{-1}$ | Inorganic 60 kg N ha$^{-1}$ | Inorganic 40 kg N ha$^{-1}$ |
| T3 | 37.5% | Organic 75 kg N ha$^{-1}$ + Inorganic 125 kg N ha$^{-1}$ | Organic 75 kg N ha$^{-1}$ + Inorganic 25 kg N ha$^{-1}$ | Inorganic 60 kg N ha$^{-1}$ | Inorganic 40 kg N ha$^{-1}$ |
| T4 | 25.0% | Organic 50 kg N ha$^{-1}$ + Inorganic 150 kg N ha$^{-1}$ | Organic 50 kg N ha$^{-1}$ + Inorganic 50 kg N ha$^{-1}$ | Inorganic 60 kg N ha$^{-1}$ | Inorganic 40 kg N ha$^{-1}$ |
| T5 | 12.5% | Organic 25 kg N ha$^{-1}$ + Inorganic 175 kg N ha$^{-1}$ | Organic 25 kg N ha$^{-1}$ + Inorganic 75 kg N ha$^{-1}$ | Inorganic 60 kg N ha$^{-1}$ | Inorganic 40 kg N ha$^{-1}$ |
| T6 | 0 | 0 | 0 | 0 | 0 |

The organic fertilizer used in the experiment was a commercial organic fertilizer developed by Gansu Dahang Agricultural Technology Company. Total N, P, and K contents in organic fertilizers were 3.3%, 1.0%, and 0.7%, respectively, and the organic matter content was >64.0%. The amount of organic fertilizer used in T2, T3, T4, and T5 treatments was 3.0, 2.3, 1.5, and 0.8 t ha$^{-1}$, respectively. For phosphorus, 150 kg P ha$^{-1}$ was applied at sowing using calcium superphosphate ($P_2O_5$ > 16.0%) to all plots. The plot area for each replicate was 37.4 m$^2$ (8.5 m × 4.4 m). The layout of the experimental blocks is shown in Figure 1.

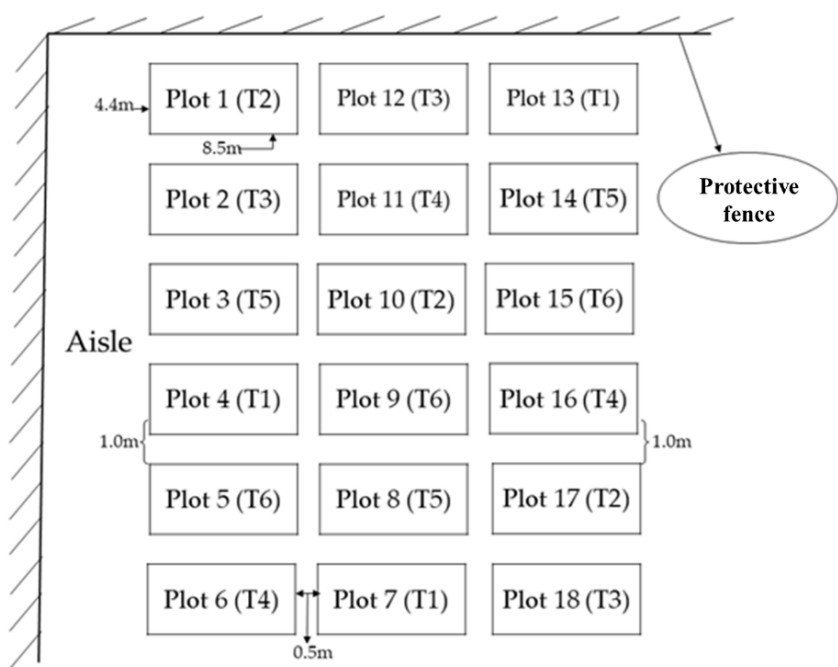

**Figure 1.** The layout of the experimental blocks.

Field Management

The alternative ridges and furrows technique was adopted for the plantation of maize (Figure 2). The ridges were mulched with a transparent polyethylene film (140 cm wide and 0.01 mm thick). The ridges were of two different sizes, i.e., wide-low ridges (70 cm in width and 15 cm in height) for field operations and narrow-high ridges (4 cm in width and 20 cm in height) for rainwater harvesting. Maize seeds were sown in furrows between two ridges. Two corn seeds were sown in each hole at a plant-to-plant spacing of 35 cm, and the seedlings were thinned. A seepage hole was dug every 1 m to ensure the effective infiltration of precipitation. No irrigation was performed during the experiment. A nitrogen

fertilizer was applied at a depth of 6–7 cm. The disease, insect and grass damage, and other management were the same as those in the general high-yield field. Corn was harvested at ripening.

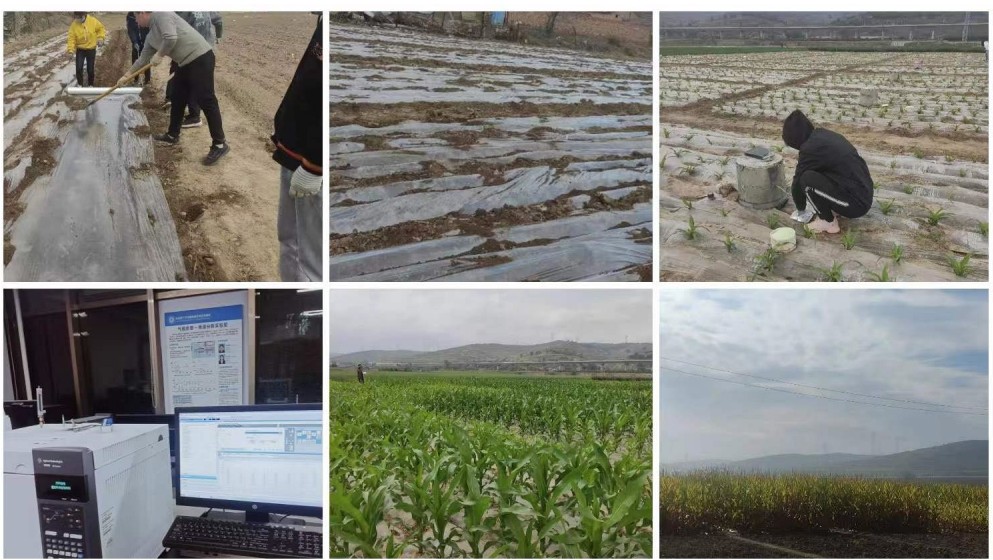

**Figure 2.** Photos of the experiment.

*2.3. Sample Collection and Measurement*

2.3.1. Gas Flux

Nitrous oxide was collected in a static closed chamber (Figure 3). The chamber is cylindrical with an outer diameter of 38 cm, a height of 35 cm, an inner diameter of 36.5 cm, and is made of a 1 mm-thick stainless-steel plate. It was buried at the center of each plot and was not moved throughout the year [7]. The box as coated with reflective aluminum foil thermal insulation films, and the top of the box was equipped with a round plug for inserting a thermometer to read the temperature in the box. A plastic fan with a width of 10 cm was installed on the side wall of the box for mixing the gas. There is an air hole in the side wall of the box, and a rubber tube is connected to the syringe for sampling. The gas samples were collected once or twice a month after maize sowing in 2020 and 2021 [23]. Samples were analyzed using an Agilent 7080B gas chromatograph (Agilent 7080B, Santa Clara, CA, USA) under the following chromatographic conditions: PoraparkQ15m $\times$ 0.53mm $\times$ 25µm, injection port at 150 °C, split injection, ECD detector, detection temperature at 300 °C, column temperature at 45 °C, column flow rate of 3.3 mL min$^{-1}$, and the carrier gas had high-purity $N_2$.

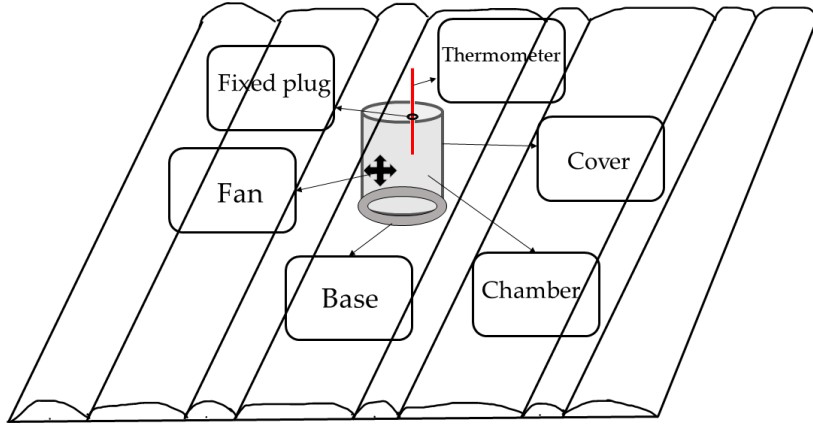

**Figure 3.** Location and design of a static closed chamber for gas sampling in the experimental plot.

The N2O fluxes (NF, mg m$^{-2}$ h$^{-1}$) were calculated using Equation (1):

$$NF = \frac{273}{273 + T} \times \frac{44}{22.4} \times 60 \times 10^{-3} \times h \times \frac{dc}{dt} \tag{1}$$

where T ($^\circ$C) is the air temperature, 44 is the molecular weight of N$_2$O, 22.4 (L mol$^{-1}$) is the molecular volume at 101 kPa, $60 \times 10^{-3}$ is a conversion factor, h is the height of the chamber, and dc/dt is the change in N$_2$O concentration (c) per unit of time (t).

N$_2$O gross emissions (NE, Kg ha$^{-1}$) were calculated using Equation (2):

$$NE = \sum [\frac{NF_{i+1} + NF_i}{2} \times (t_{i+1} - t_i) \times 24 \times 10^{-2}] \tag{2}$$

where *i + 1* and *i* are the dates of the last and current measurements, respectively, and t is the number of days after sowing.

### 2.3.2. Soil Sample

Five subsamples of soil were collected from each replication with an auger (3.4 cm in diameter) from a depth of 20 cm. Subsamples were mixed to make a single sample and were passed through a 1 mm sieve. Then, DNA was extracted from the soil using OMEGA kits according to the instructions. The quality and concentration of the extracted DNA were checked by a spectrophotometer (Nanodrop; PeqLab, Erlangen, Germany), and samples were stored at $-80$ $^\circ$C. The number of target microbial species was calculated by using fluorescence quantification by labeling and tracking PCR products with fluorescent dyes or fluorescent labeled specific probes. High-throughput sequencing technology was used to detect a large number of nucleic acid molecules simultaneously and to decrypt the genetic code of the genome of a target species.

Highly variable regions of bacterial 16S rRNA genes of ammonia-oxidizing archaea and ammonia-oxidizing bacteria were amplified by using barcoded primers with sequence STAATGGTCTGGCTTAGACG/GCGGCCATCCATCTGTATGT and GGGGTTTCTACTG-GTGGT/CCCCTCKGSAAAGCCTTCTTC, respectively. The reaction solution for PCR consisted of a 15 $\mu$L qPCR mix, 2 $\mu$L Mg$^{2+}$, 0.5 $\mu$L forward primer, 0.5 $\mu$L reverse primer, 2 $\mu$L template, and then a 30 $\mu$L volume was maintained by adding ddH$_2$O. Thermal cycling consisted of an initial denaturation at 95 $^\circ$C for 3 min, followed by 40 cycles of denaturation at 95 $^\circ$C for 30 s, annealing at 50 $^\circ$C for 30 s, and elongation at 72 $^\circ$C for 30 s. Each purified PCR product was amplified using Illumina MiSeq (Shanghai Biozeron Biotechnology Co., Ltd., Pudong, Shanghai, China).

The raw data obtained from all samples based on a barcode were filtered for the quality of the reads, and pairs of reads were spliced into a single sequence based on the overlap between PE reads. According to the positive and negative barcode and primer orientation, the sequence orientation was corrected, and chimeras were removed. DNA fragments were merged using FLASH (version 1.2.7). Sequence analyses were performed with the UPARSE (version 7.1) software package using UPARSE-OTU and UPARSE-OTUref algorithms. Alpha-diversity analysis was performed using MOTHUR software (version 1.44.1). The sequencing reads were submitted to the Sequence Read Archive (SRA) at the National Center for Biotechnology Information (NBCI) under the BioProject PRJNA821646.

### 2.3.3. Above-Ground Biomass

Above-ground biomass was determined as the sum of grain yield and straw weight. At maturity, 30 plants were randomly harvested from each plot, and their roots were separated and weighed.

### 2.4. Data Analysis

The data of biomass, soil physicochemical properties, N$_2$O total emission, the copy number of AOA and AOB genes, and diversity indices of amoA genes were subjected to analyses of variance (ANOVA) using SPSS software (v.20.0, IBM Corp., Chicago, IL, USA).

To investigate complex co-occurrence patterns in soil AOA and AOB communities, OTUs were pooled and filtered. The co-occurrence network's visualization was conducted using Gephi software, and modules were defined as clusters of closely interconnected nodes. A Bray–Curtis pairwise analysis was conducted using the R program. A random forest algorithm (rfPermute package) was used to predict nitrous oxide emissions. Structural equation modeling (SEM) based on maximum likelihood estimation was performed on Amos 22.0 (SPSS, Chicago, IL, USA) statistical software. The ratio of chi-squared to *p*-value (Normed chi-square, NC) and comparative fit index (CFI) was used to evaluate the overall goodness of the model. The following judgment criteria were used: NC values between 1 and 3 are preferable, and those less than 5 are acceptable. CFI values greater than 0.9 are preferable. All data were expressed in terms of mean $\pm$ standard deviation based on three replicates.

## 3. Results

### 3.1. $N_2O$ Emissions, Grain Yield, and Above-Ground Biomass

$N_2O$ emission fluxes were predominantly higher for fertilizer-treated plots than for the control (without fertilizer) treatment throughout the maize reproductive stage (Figure 4). After applying a basal fertilizer, $N_2O$ emission fluxes showed a gradual decrease near the control in both years, and the fluxes increased in June and July and then decreased again.

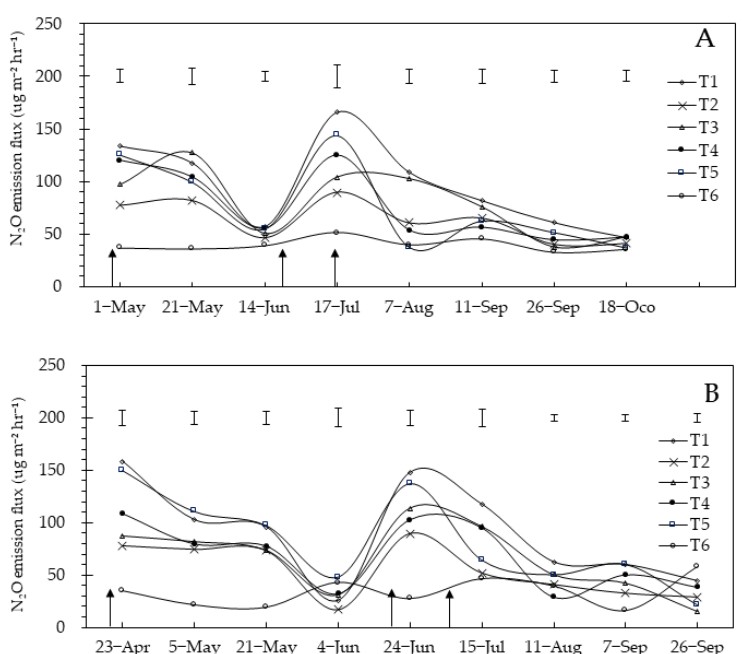

**Figure 4. A and B represent the years 2020 and 2021 respectively.** The dynamic changes of nitrous oxide emissions during the entire growth period of maize. The vertical bars represent the least significant difference (LSD) at *p* < 0.05. Arrows represent fertilization. T1: inorganic fertilizer; T2: 50.0% organic fertilizer; T3: 37.5% organic fertilizer; T4: 25.0% organic fertilizer; T5: 12.5% organic fertilizer; T6: no fertilizer.

The total $N_2O$ emission for the fertilized plots was significantly higher in both years than in the control (Figure 5A). Substituting inorganic N with organic fertilizers reduced the total emission of $N_2O$, and increased reductions in $N_2O$ were observed by substituting a higher portion of inorganic N, i.e., 50.0% substitution (T2), followed by 37.5% substitution (T3). Replacing inorganic fertilizers with organic fertilizers did not show any significant reductions in grain yield (Figure 5B) or aboveground biomass (Figure 5C).

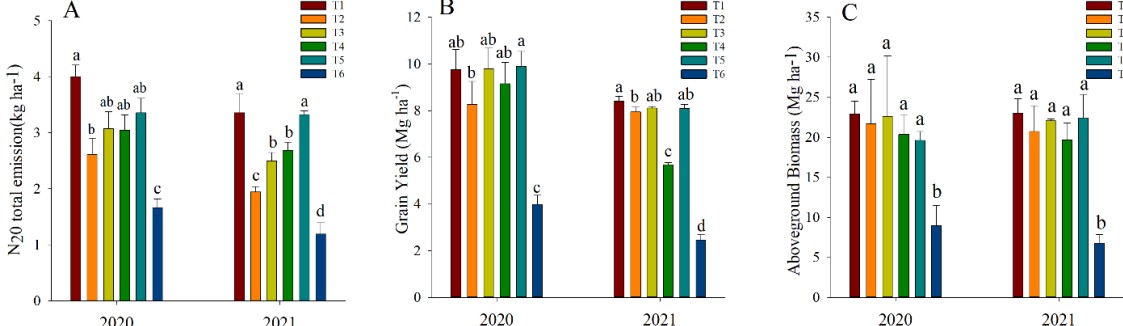

**Figure 5.** Total nitrous oxide emissions of the entire growth period (**A**) and grain yield (**B**) aboveground biomass (**C**). The different alphabets (a, b, c and d) show significant difference at $p < 0.05$. T1: inorganic fertilizer; T2: 50.0% organic fertilizer; T3: 37.5% organic fertilizer; T4: 25.0% organic fertilizer; T5: 12.5% organic fertilizer; T6: no fertilizer.

### 3.2. Soil Physiochemical Properties

It can be seen from Table 3 that the addition of different proportions of organic fertilizer can increase soil pH and organic carbon contents to different degrees and improve the soil water environment of dryland farmland compared with the 100% application of inorganic fertilizer. The soil pH by 50% substitution (T2) was significantly higher than the 12.5% substitution (T5). Adding a 37.5% organic fertilizer (T3) increased the pH from 8.01 to 8.09, soil water content from 8.2 to 9.4%, and SOC by 2.5% compared to the inorganic fertilizer. However, TN and $NO_3^- - N$ contents by the addition of the 37.5% organic fertilizer were 17.2% and 19.0% less than the inorganic fertilizer. Substituting different portions of inorganic fertilizers with organic fertilizer did not affect $NH4^+ - N$ contents compared to using the inorganic fertilizer alone. Adding different proportions of organic fertilizer did not affect the TN content, but the soil moisture content of dryland significantly improved with the increase in substitution ratios than that of the inorganic fertilizer alone.

**Table 3.** Soil physicochemical properties at the flowering stage of maize supplied with different ratios of organic and inorganic fertilizers.

| Treatment | pH | TN g kg$^{-1}$ | SOC g kg$^{-1}$ | $NO_3^- - N$ mg kg$^{-1}$ | $NH_4^+ - N$ mg kg$^{-1}$ | Moisture % |
|---|---|---|---|---|---|---|
| T1 | 8.01 ± 0.06 [c] | 1.16 ± 0.01 [a] | 9.22 ± 0.47 [b] | 25.43 ± 1.25 [a] | 17.16 ± 2.01 [a] | 8.21 ± 0.77 [d] |
| T2 | 8.16 ± 0.07 [b] | 0.99 ± 0.07 [b] | 9.60 ± 0.15 [a] | 20.34 ± 1.59 [b] | 14.33 ± 1.09 [ab] | 11.07 ± 0.64 [b] |
| T3 | 8.09 ± 0.09 [bc] | 0.96 ± 0.07 [b] | 9.45 ± 0.31 [ab] | 20.59 ± 2.17 [b] | 15.33 ± 1.57 [ab] | 9.44 ± 0.54 [c] |
| T4 | 8.06 ± 0.02 [bc] | 1.01 ± 0.05 [b] | 9.28 ± 0.20 [ab] | 22.58 ± 1.96 [ab] | 15.89 ± 1.32 [ab] | 8.41 ± 0.22 [d] |
| T5 | 8.02 ± 0.03 [c] | 0.97 ± 0.06 [b] | 9.40 ± 0.34 [ab] | 22.60 ± 1.06 [ab] | 14.79 ± 1.91 [ab] | 8.31 ± 0.43 [d] |
| T6 | 8.30 ± 0.04 [a] | 0.83 ± 0.02 [c] | 8.67 ± 0.13 [c] | 16.22 ± 0.98 [c] | 14.19 ± 0.66 [b] | 13.01 ± 0.51 [a] |

Values are expressed as a mean with standard error. Means in a column with a common alphabet (a) as superscript indicate no significant difference at $p < 0.05$ while the different alphabets (a, b, c and d) show significant difference at $p < 0.05$. Abbreviations: total nitrogen (TN); soil organic carbon (SOC); nitrate nitrogen ($NO_3^- - N$); ammonia nitrogen ($NH_4^+ - N$). T1: inorganic fertilizer; T2: 50.0% organic fertilizer; T3: 37.5% organic fertilizer; T4: 25.0% organic fertilizer; T5: 12.5% organic fertilizer; T6: no fertilizer.

### 3.3. Abundance and Diversity of Ammonia Oxidizers

There was no significant effect of inorganic and organic fertilizers on the gene abundance of AOA, while they differently affected AOB gene's abundance, with copy numbers ranging from $3.8 \times 10^6$ to $8.9 \times 10^6$ (Figure 6). Compared with inorganic fertilizer application, 37.5% of organic fertilizer replacement treatment (T3) reduced the copy number of AOB but did not reach a significant level. The AOB copy number of organic fertilizer replacement treatment (T2) was reduced by 50.0% compared with T1, while the AOB copy number of organic fertilizer replacement treatments (T2, T3, T4, and T5) was not significantly different.

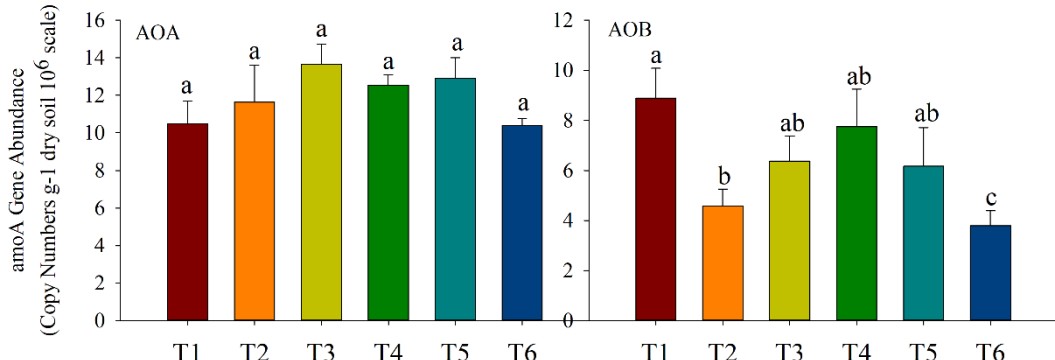

**Figure 6.** amoA gene copy numbers of AOA and AOB in soil under different substitution ratios. The different uppercase letters indicate significant differences among soil amendments, with error bars that represent the standard deviation ($p < 0.05$, Duncan's multiple-range test). T1: inorganic fertilizer; T2: 50.0% organic fertilizer; T3: 37.5% organic fertilizer; T4: 25.0% organic fertilizer; T5: 12.5% organic fertilizer; T6: no fertilizer.

OTUs and Chao1 indices showed that fertilizer treatments significantly increased AOA and AOB richness compared to the control, but the diversity of AOA did not differ significantly between fertilizer treatments (Table 4). The organic fertilizer's substitution significantly reduced OTUs' number of AOB compared to urea alone. The number of OTUs significantly increased in the organic fertilizer replacement treatment (T3) compared with other organic fertilizer replacement treatments (T4, T5). The Shannon index of AOA was not affected but that of AOB showed a significant increase by the fertilizer treatments, but the effect of organic fertilizer was non-significant compared to inorganic fertilizers.

**Table 4.** Diversity indices of ammonia-oxidizing archaea (AOA) and ammonia-oxidizing bacteria (AOB) at a similarity level of 99% under different substitution ratios.

| Treatment | AOA | | | AOB | | |
|---|---|---|---|---|---|---|
| | OTUs | Chao1 | Shannon | OTUs | Chao1 | Shannon |
| T1 | 731.2 ± 6.1 [a] | 735.6 ± 19.7 [a] | 5.1 ± 0.27 [a] | 1381.1 ± 26.7 [a] | 1378.0 ± 48.0 [a] | 6.8 ± 0.27 [a] |
| T2 | 734.2 ± 72.2 [a] | 732.1 ± 77.0 [a] | 5.0 ± 0.17 [a] | 1214.2 ± 47.5 [bc] | 1200.5 ± 32.82 [c] | 6.5 ± 0.37 [a] |
| T3 | 697.1 ± 30.7 [ab] | 682.8 ± 18.8 [a] | 5.0 ± 0.32 [a] | 1245.8 ± 23.7 [b] | 1243.7 ± 32.97 [b] | 6.6 ± 0.27 [a] |
| T4 | 697.9 ± 29.1 [ab] | 703.4 ± 19.9 [a] | 5.1 ± 0.20 [a] | 1137.3 ± 29.76 [c] | 1242.0 ± 46.98 [b] | 6.1 ± 0.73 [a] |
| T5 | 721.1 ± 35.7 [a] | 702.3 ± 28.4 [a] | 5.2 ± 0.31 [a] | 1166.7 ± 69.67 [c] | 1369.7 ± 30.60 [a] | 6.8 ± 0.29 [a] |
| T6 | 633.3 ± 14.0 [b] | 616.7 ± 17.2 [b] | 5.1 ± 0.11 [a] | 915.4 ± 27.3 [d] | 715.9 ± 16.32 [d] | 5.2 ± 0.31 [b] |

Data were presented as mean with standard error. Means in a column with a common alphabet (a) as superscript indicate no significant difference at $p < 0.05$ while the different alphabets (a, b, c and d) show significant difference at $p < 0.05$. T1: inorganic fertilizer; T2: 50.0% organic fertilizer; T3: 37.5% organic fertilizer; T4: 25.0% organic fertilizer; T5: 12.5% organic fertilizer; T6: no fertilizer.

### 3.4. The Co-Occurrence Network of Ammonia Oxidizers

The correlation network showed that AOA in soil essentially belongs to the genus *Nitrososphaera* and AOB to the genus *Nitrosospira* (Figure 7). The network graph of AOA contained 178 nodes and 761 edges, with an average degree of 14.74, a network diameter of 8, modularity of 0.47, an average clustering coefficient of 0.581, and an average path length of 3.31 edges. The positive synergistic relationship between clusters was greater than the negative antagonistic relationship and formed 99.5% of the total correlation (Figure 7A). The network structure of AOB had 84 nodes with 399 edges, a modularity of 0.88, and an average clustering coefficient of 0.55 (Figure 7B). The positive synergistic relationships were greater than the negative antagonistic relationships between clusters, contributing to 85.2% of the relationships.

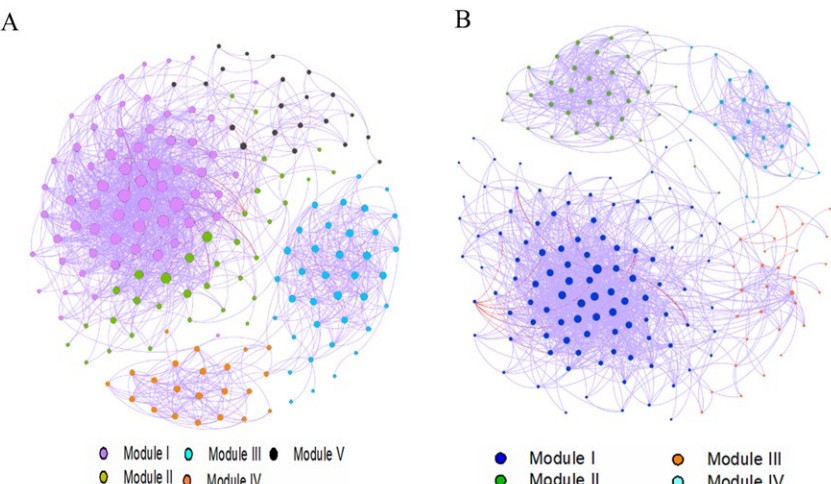

**Figure 7.** Network analysis of (**A**) AOA and (**B**) AOB OTUs modules. Modules were labelled with different colors in each network. The size of the OTU nodes indicates their degrees, and the purple line indicates a positive correlation, while the red line illustrates a negative correlation.

The correlation was performed using module data obtained from the network analysis and the soil's physicochemical characteristics, $N_2O$ emissions, and the biomass of maize (Figure 8). Among the five modules obtained from the network analysis of AOA, the OTUs in the M3 module are significantly and positively correlated with soil pH and soil water content, while they were negatively correlated with soil organic carbon, $N_2O$ emissions, and aboveground biomass (Figure 8A). The abundance, richness, and composition of the AOB community were significantly correlated with soil physicochemical indicators, except for $NH_4^+-N$ (Figure 8B). The abundance and richness of the AOB community were positively correlated with total $N_2O$ emissions and aboveground biomass, whereas AOB composition was negatively correlated with $N_2O$ emissions and aboveground biomass. Among the four modules, the OTUs in M2 were the most relevant to the indicators obtained from the analysis of the AOB network.

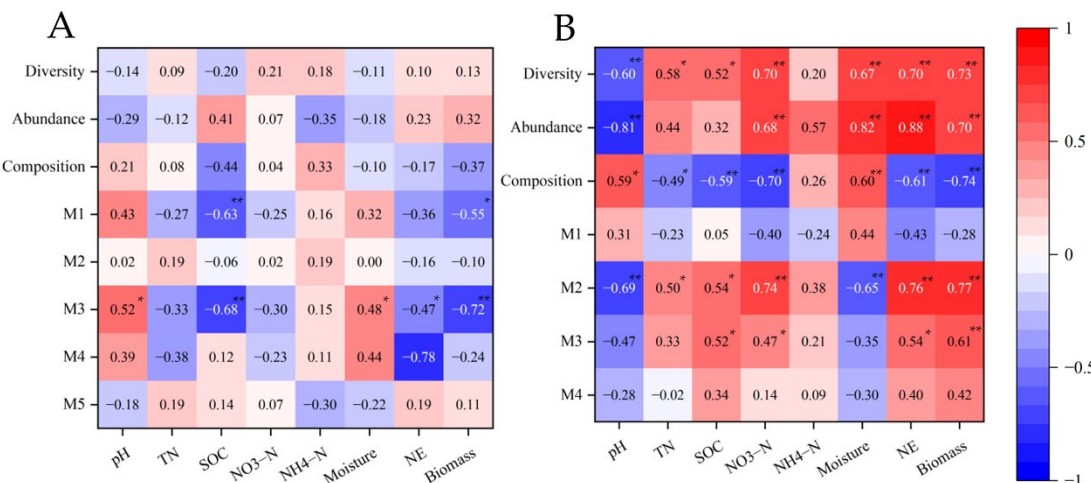

**Figure 8. A and B represent the AOA and AOB respectively.** Correlation analysis of microbial indicators on physicochemical properties of soil and $N_2O$ emissions and above-ground biomass. Red color represents positive correlations and blue represents negative correlations. The asterisk symbol represents statistical significance, where * stands for $p < 0.05$ and ** stands for $p < 0.01$. Abbreviations: total nitrogen (TN); soil water content (WC); soil organic carbon (SOC); nitrate nitrogen ($NO_3-N$); ammonia nitrogen ($NH_4-N$); $N_2O$ emission (NE).

### 3.5. Quantitative Analysis of Soil Properties and Microorganisms on N₂O Emissions

To verify the relationship between N$_2$O emission, soil physicochemical properties, and the diversity of AOA and AOB, we calculated the contribution of each factor to N$_2$O emissions by using a random forest algorithm (Figure 9). The diversity, abundance, and composition of AOB and the contribution of OTUs species in M2 to N$_2$O emission were 6.5%, 10.8%, 9.5%, and 11.2%, respectively.

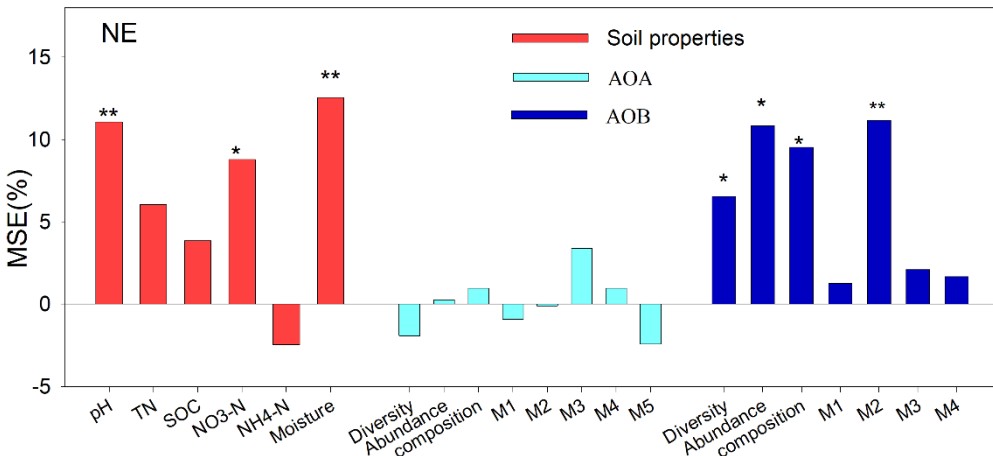

**Figure 9.** Association of nitrous oxide gas emissions with soil properties and ammonia oxidizing archaea and bacteria by a random forest algorithm. Significant predictors were chosen to perform the SEM analysis. This includes the following: total nitrogen (TN); soil organic carbon (SOC); nitrate nitrogen (NO$_3^-$−N); ammonia nitrogen (NH$_4^+$−N); diversity (Shannon index); abundance (copy numbers); composition (first principal coordinates, PC1); and seven module eigengenes. * $p < 0.05$; ** $p < 0.01$.

Structural equation modeling (SEM) was performed from the results of the random forest algorithm to hypothesize the contribution of indicators relative to N$_2$O emissions (Figure 10). SEM indicates that substituting inorganic fertilizers with different proportions of organic fertilizer changed the structure and components of the AOB community by affecting soil pH, NO$_3^-$−N, and moisture contents, which ultimately changed the total amount of N$_2$O emissions, and its operational parameters met modeling requirements.

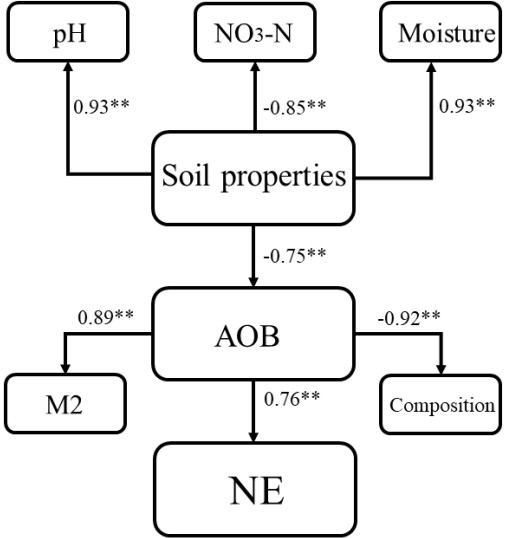

**Figure 10.** Structural equation modelling of ammonia-oxidizing bacterial communities with variability in soil properties. Significant levels are denoted: ** $p < 0.01$. Standardized total effects (direct and indirect effects) calculated by SEMs are displayed.

The relative abundance of key OTUs in M2 is shown under different treatments (Figure 11). AOB1, AOB50, AOB52, AOB79, AOB83, AOB91, AOB97, and AOB104 were more significantly expressed. The relative abundance of OTUs was higher in the no-fertilizer than in the fertilizer treatment, and the highest expression was found in the 50.0% substitution with an organic fertilizer.

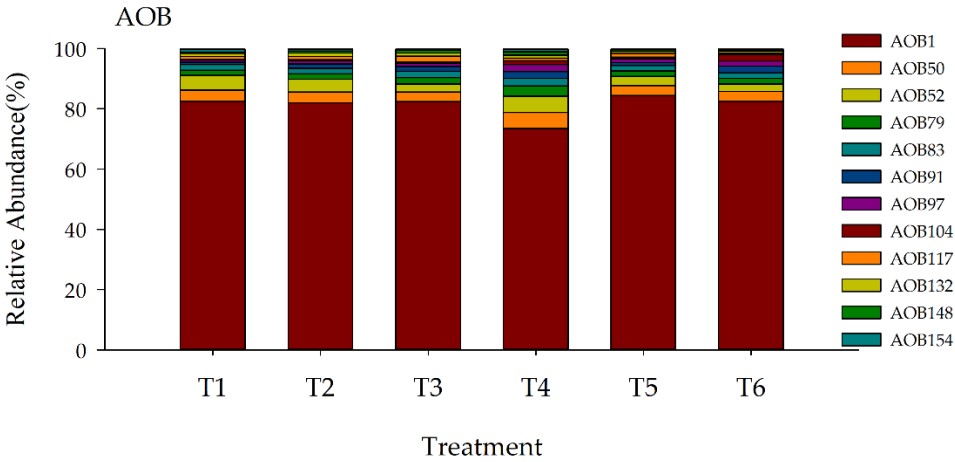

**Figure 11.** Relative abundance of selected Genus cluster in the M2 module of AOB among dominant 98 OTUs in response to organic fertilizer substitution. Twelve OTUs were selected to account for more than 1% of each treatment. T1: inorganic fertilizer; T2: 50.0% organic fertilizer; T3: 37.5% organic fertilizer; T4: 25.0% organic fertilizer; T5: 12.5% organic fertilizer; T6: no fertilizer.

## 4. Discussion

In the current study, the effect of replacing different rates of inorganic N fertilizer (urea) with organic fertilizer was studied on the diversity of ammonia-oxidizing bacteria (AOB), archaebacteria (AOB), and $N_2O$ emission fluxes. Nitrogen fertilizers shifted the AOB community. The AOA copy number of each treatment ranged from $10.4 \times 10^6$ to $13.7 \times 10^6$, with no significant differences between fertilizer treatments. Similarly, most previous studies showed no significant effects of N fertilizers on the abundance of AOA communities in alkaline soils [24–26]. However, the AOB gene copy number increased by inorganic N fertilizer (urea) and decreased by substituting urea with organic N fertilizers compared to urea alone, and a maximum decrease was observed under 50% substitution (Figure 6). This showed that organic N fertilizers were less disturbed the AOB community. This may be the reason why the addition of organic fertilizer in this experiment can reduce the emission of nitrous oxide in farmland [27–29]. The effects of different proportions of organic fertilizer replacement treatments on AOA and AOB were different than those of inorganic fertilizer alone. Substituting urea with organic fertilizer showed no significant differences in AOA community abundance compared to urea applications, but AOB community abundance was significantly reduced by 50.0% substitution with organic fertilizers compared to urea alone. However, the substitution ratio of 37.5%, 25%, and 12.5% had no significant changes compared with the inorganic fertilizer alone, which may be caused by a quantitative variation in the high substitution ratio.

The 12.5% and 25.0% substitution of urea with organic fertilizer reduced the OTUs of AOB more than the 37.5% and 50.0% substitutions. However, 37.5% substitution showed a balance between a reduction in $N_2O$ emissions and maize growth. This suggests that the number of functional microbial OTUs may not fully reveal $N_2O$ emission patterns between different fertilizer treatments. Therefore, further data mining is required to investigate how microbial community structures may affect $N_2O$ emissions under organic and inorganic fertilizers.

In this trial, $N_2O$ emissions were monitored for two consecutive years during the entire maize growth period. Different organic fertilizer substitution rates reduced $N_2O$ emissions to different degrees. The 37.5% substitution of inorganic N fertilizer reduced

the $N_2O$ emission compared to inorganic fertilizers in both years, and a more pronounced reduction in $N_2O$ emissions was observed in the dry year (2021) than in the wet year (2020).

It has been widely reported that the soil moisture's status is a significant variable explaining changes in $N_2O$ emissions [30,31]. The results of this study showed that the inorganic fertilizer reduced soil moisture, while 50.0 and 37.5% substitutions of inorganic fertilizer had higher soil moisture compared to inorganic fertilizer applications (Table 3). Soil moisture significantly correlated with the OTUs contained in M3 obtained from the AOA network analysis (Figure 8) and with the diversity, abundance, composition, and OTUs within the M2 of the AOB. Numerous studies have also shown that the soil environment affects the community structure of AOA and AOB [28,32,33]. In this study, the correlation analysis of pH, TN, SOC, $NO_3^- - N$, and $NH_4^+ - N$ indicators revealed that pH, SOC, and $NO_3^- - N$ were as close as that between moisture and microbial community structure indicators. The random forest algorithm analysis showed that pH, $NO_3^- - N$, soil moisture, and OTUs contained in M2 for AOB were the most critical factors involved in $N_2O$ emissions. The structural equation model systematically explained the positive correlation between organic fertilizer inputs and pH and moisture content and the negative correlation between organic fertilizer inputs and soil $NO_3^- - N$. In this study, ammonium N and soil organic matter content did not respond well to organic fertilizer inputs, which may be related to sampling times or soil texture [34–37]. There was a highly negative correlation between changes in soil physicochemical properties and AOB, and the OTUs obtained from the M2 module explained 89% of the function of AOB. A positive value for the pathway interpretation coefficient of the AOB fraction on $N_2O$ emissions and higher expression of OTUs in M2 under the no-fertilizer treatment than in the fertilizer treatment indicated the role of AOB abundance in $N_2O$ emission. Furthermore, there was a higher relative gene abundance of AOB under inorganic fertilizers than in the different rates of organic fertilizer substitution, which demonstrated their roles in $N_2O$ emissions.

## 5. Conclusions

In this study, the highest $N_2O$ emission was observed when using inorganic fertilizers, and AOB contributed more $N_2O$ emissions than AOA. Replacing inorganic N fertilizers with different rates of organic fertilizers reduced the flux of $N_2O$ compared to inorganic fertilizers alone. By combining maize grain yields and biomass, it is most appropriate to replace inorganic fertilizers with a ratio of 37.5%. The addition of organic fertilizers reduced the AOB copy number by affecting the pH, soil moisture, and $NO3^- - N$ content. Nitrososphaera (AOA) and Nitrosospira (AOB) were the most dominant genus. Observing the effect of different organic fertilizer replacement rates on $N_2O$ emissions and changes in nitrifying bacterial communities is essential for the development of more sustainable fertilizer practices.

**Author Contributions:** Conceptualization, L.X. and L.L.; methodology, L.L. and L.X.; validation, L.X.; resources, J.X. and J.W.; data curation, L.X. and J.W.; writing—original draft preparation, L.X.; writing—review and editing, C.D., Y.Z., S.A. and L.L.; visualization, L.X.; supervision, L.L.; project administration, J.X. All authors have read and agreed to the published version of the manuscript.

**Funding:** The research was supported by the National Natural Science Foundation of China (31761143004); and the Innovation Star Project for Excellent Graduate Student of Gansu Province Educational Department (2021CXZX-370).

**Data Availability Statement:** Not applicable.

**Acknowledgments:** We sincerely thank the undergraduate and graduate students of the Rain Agriculture Experimental Station of Gansu Agricultural University for their excellent technical assistance in field sampling and laboratory testing.

**Conflicts of Interest:** The authors declare no conflict of interest.

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
