# Peer review of "Substituting Inorganic Fertilizers with Organic Amendment Reduced Nitrous Oxide Emissions by Affecting Nitrifiers’ Microbial Community"

_land, doi:10.3390/land11101702_

Round 1

Reviewer 1 Report

The manuscript deals with the effects of organic substitution on soil properties, N2O emissions, and microbes in a semiarid maize field. The topic is of interest and fits the journal. I have several suggestions that the authors may reconsider.

Major issues:

1. The TITLE highlights denitrifiers while the research studies AOB and AOA (nitrifiers)???

2. The gas flux measurement is not clear. How the chambers were set up? Is there any protocol to ensure that the sampling area is representative at the field scale? See Cai et al. 2012.

Yanjiang Cai, Weixin Ding, Jiafa Luo, Spatial variation of nitrous oxide emission between interrow soil and interrow plus row soil in a long-term maize cultivated sandy loam soil, Geoderma 2012, 181–182, 2-10, https://doi.org/10.1016/j.geoderma.2012.03.005.

3. I am interested in the comparison of the observations between the substitution ratios rather than the fertilized treatments to the unfertilized treatment. The authors may reconsider their Result and Discussion.

Other things:

1. in Abstract, organic fertilizers, organic manure, soil fertility, a reasonable proportion (L25) are not clear. The Control (L17) treatment is not defined. The study reported above ground biomass rather than grain yields (L15,18,20 and other places in the text). Please clarify.

2. L39, 41, please update according to the latest IPCC report.

3. How the field are managed (tillage, irrigation, crop residues, ......) is not clear.

4. Fig. 1, please indicate fertilization dates.

Author Response

Q1. The TITLE highlights denitrifiers while the research studies AOB and AOA (nitrifiers)???

Answer: The word denitrifiers' in the title has been changed to nitrifiers'. Thanks to the expert for hard work.

Q2. The gas flux measurement is not clear. How the chambers were set up? Is there any protocol to ensure that the sampling area is representative at the field scale? See Cai et al. 2012.

Answer: Thanks to the expert for professional advice. Add content under heading 2.3.1 following expert advice: Nitrous oxide is collected by static closed chamber technique. We have added Figure 3 showing the setup of the chamber and have added the details

Figure 3. Location of static closed-chamber deployment for gas sampling in the center of each plot and the specific design of the chamber.

Q3. I am interested in the comparison of the observations between the substitution ratios rather than the fertilized treatments to the unfertilized treatment. The authors may reconsider their Result and Discussion.

Answer: According to expert advice, changed the Result and compared the changes within different ratios of substitution, where the effect was significantly different among different substitution rates.

Other things:

Q1. in Abstract, organic fertilizers, organic manure, soil fertility, a reasonable proportion (L25) are not clear. The Control (L17) treatment is not defined. The study reported above ground biomass rather than grain yields (L15,18,20 and other places in the text). Please clarify.

Answer: Based on expert opinion, in Abstract, organic fertilizer has been explained in the commercial organic fertilizer. The expression is changed from “... 0, 50, 37.5 25, and 12.5% were replaced with organic fertilizers…” to “… 0, 50, 37.5 25, and 12.5% were replaced with organic fertilizers which were commercial organic fertilizer”. Supplementary noted Control from “Results showed that the maximum N2O emission was by 100% chemical fertilizer and the lowest at control” to “Results showed that the maximum N2O emission was by 100% chemical fertilizer and the lowest at control (no fertilizer)”. The expression of soil fertility changed from “Substituting chemical fertilizer with organic manure not only reduced N2O emissions but also improved soil fertility and ensures grain yield and biomass.” To “Substituting chemical fertilizer with organic manure not only reduced N2O emissions but also improved soil organic carbon content, moisture, stable pH and ensures grain yield and biomass.”. “a reasonable proportion” was made clear to “… replacing a reasonable proportion-37.5% of chemical fertilizer …”. There is no doubt about the importance of grain yield index of field maize, hence, the grain yield should be considered when selecting the optimal treatment, and the "Xianyu 335" studied in this paper, as a kind of corn variety with both grain and feed. The use of biomass indicators in the analysis is not only of experimental significance, but also more fit the actual production situation.

Q2. L39, 41, please update according to the latest IPCC report.

Answer: According to the latest IPCC report, the original content has been updated from ” Almost 75% of N2O is emitted from agricultural activities” to “Approximately 80% of N2O is emitted from agricultural activities”. Meanwhile, The original reference has also been replaced as “Pörtner, H. O.; Roberts, D, C.; Adams, H.; Adler, C.; Aldunce, P. Ali, E.; Fischlin, A. Climate change 2022: Impacts, adaptation and vulnerability. IPCC Sixth Assessment Report. 2022, doi.org/10.1111/padr.12497.” We gratitude to expert's hard work.

Q3. How the field are managed (tillage, irrigation, crop residues, ......) is not clear.

Answer: Thanks for the expert's opinion, supplementary title 2.2.1. Field management, and the content as follow: In order to prevent soil moisture loss in winter, after corn harvest in 2019, straw was harvested and removed from the field without uncovering the film, and the soil was no-tillage. After the soil thawed in 2020, the residual film was recovered manually, and the base fertilizer was applied according to the fertilization scheme. Rotary tillage and furrow raising with whole film on double ridges were used to raise large and small ridges (large ridges: ridge height 15 cm, ridge width 70 cm; Small ridge: ridge height 20 cm, ridge width 40 cm). A white plastic film 140 cm wide and 0.01 mm thick was used to cover the whole surface, and a seepage hole was dug every 1m in the trench to ensure the effective infiltration of precipitation (no irrigation). Two corn seeds were sown in each hole at a spacing of 35cm in the furrow, and the seedlings were thinning in time. Nitrogen fertilizer was applied topdressing between the two corn plants at jointing stage and pre-tasseling stage with a depth of 6-7cm. The disease, insect and grass damage and other management were the same as those in the general high-yield field. Corn was harvested according to field ripening, and the plan for 2021 was the same as that for 2020.

Q4. Fig. 1, please indicate fertilization dates.

Answer: Thanks for the expert's professional guidance. We have already indicated fertilization dates by arrows in Fig.1. as follow and added “Arrows represent fertilization.” in the note.

Reviewer 2 Report

See attached file

Author Response

Q1. The authors must rewrite the article following the rules of the journal, especially regarding the way of citing the authors and therefore the references section.

Answer: Thanks for the expert's careful guidance. The font of the table is changed from “Times New Roman” to “Palatino Linotype” according to the format of published papers in this journal. Change the way of citing the authors from “(Jackson and Forster, 2021; Khan and Paul, 2021)” to “[1,2]” or from “(Dick et al. 2008; Morimoto et al. 2009; Barton et al. 2013)” to “[11–13]”. Examples of references are as follows:

  1. Ashiq, W.; Ghimire, U.; Vasava, H; Dunfield, K.; Wagner-Riddle, C.; Daggupati, P.; Biswas, A. Identifying hotspots and representative monitoring locations of field scale N2O emissions from agricultural soils: A time stability analysis[J]. Science of The Total Environment. 2021, 788: 147955, doi.org/10.1016/j.scitotenv.2021.147955.
  2. Chirinda, N., Trujillo, C., Loaiza, S., Salazar, S., Luna, J., Tong Encinas, L.A., Becerra López Lavalle, L.A., Tran, T. Nitrous oxide emissions from cassava fields amended with organic and inorganic fertilizers[J]. Soil Use and Management. 2021, 37(2): 257-263, doi.org/10.1111/sum.12696.

Q2. Line 73. The soil at the site is a calcareous cambisol (FAO, 1990). Please provide a more recent version of soil classification.

Answer: Based on expert opinion, we have replaced the previously mentioned reference (FAO. Soil Map of the World: Revised Legend; World Soil Resources Report 60; Food and Agriculture Organization of the United Nations: Rome, Italy, 1990.) with (NBS (National Bureau of Statistics of China). China statistics yearbook. 2016. http://www.s-tats.gov.cn/tjsj/ndsj/2016/indexch.htm.). Thanks to the expert for hard work.

Q3. Line 111. Then DNA was extracted from the soil using OMEGA kits according to the instructions. Starting from my limited knowledge on this subject, I do not find the interest or what it contributes to the experiment. Therefore, I would ask you to study its inclusion or not. And it is left as it is, to clarify what these analyzes contribute to the study.

Answer: Thanks for the expert's professional guidance. We changed the original content to “Then DNA was extracted from the soil using OMEGA kits according to the instructions which was operated by Biozeron corporation.” Meanwhile, we added “Fluorescence quantification refers to the labeling and tracking of PCR products by fluorescent dyes or fluorescent labeled specific probes, which can quantitatively calculate the number of target microbial species. High-throughput sequencing technology can detect a large number of nucleic acid molecules simultaneously and decrypt the genetic code of the genome of a target species.” to clarify what these analyzes contribute to the study.

Q4. In relation to the experimental plots, it would be convenient to offer more details on how the experimental blocks have been arranged.

Answer: According to expert's advice, the layout of the experimental blocks was added under the title 2.2. Experimental design, as shown below:

Figure 1. The layout of the experimental blocks.

Q5. Line 140. Data analysis. There should be a new section 2.4?

Answer: Thanks to the expert's careful work, a new section 2.4 has been added.

Q6. Please improved the quality of figure 8

Answer: Thanks to the expert for careful work. The figure 8 has been changed from

to

Figure 8. Relative abundance of selected Genus cluster in the M2 module of AOB among domi-nant 98 OTUs, which were significantly changed by organic fertilizer substitution. 12 OTUs were selected to account for more than 1% of each treatment. chemical fertilizer (T1); 50.0% organic fertilizer (T2), 37.5% or-ganic fertilizer (T3); 25.0% organic fertilizer (T4); 12.5% organic fertilizer (T5); no fertilizer (T6).

Q7. I suggest to include some photos of the experiment.

Answer: According to the expert's suggestion, we added the following picture under the title 2.2.1. Field management, shown as below.

Figure 2. Photos of the experiment.

Round 2

Reviewer 1 Report

The authors addressed all issues raised by reviewers. I have no further comments.

Author Response

Prof. Dr. Christine Fürst 
Editor-in-Chief

Land Journal

Dear Christine Fürst 

Thank you for your reply and your agreement to my modification. I wish you a happy life and work.

Regards

Lingling Li

College of Agronomy,

Gansu Agricultural University, Lanzhou and China

[email protected]

Reviewer 2 Report

I note that although the answers of authors are adequate to my requirements, these are not reflected in the manuscript. Maybe I can't see the modified document. Therefore, if the modifications are included in the MS, then I could consider it accepted.

Thanks for your mail. Now I can see the file with the modifications in red. The manuscript can be accepted; but on line 76 appears calcareous cambisol. Please put both in capital letters. And add according to [23].

Author Response

Prof. Dr. Christine Fürst 
Editor-in-Chief

Land Journal

Dear Christine Fürst 

Thank you for your reply and your agreement to my modification. Based on expert opinion, “on line 76 appears calcareous cambisol. Please put both in capital letters”. The expression is changed from “calcareous cambisol” to “Calcareous Cambisol”. I wish you a happy life and work.

Regards

Lingling Li

College of Agronomy,

Gansu Agricultural University, Lanzhou and China

[email protected]
